# DNA Demethylation of Myogenic Genes May Contribute to Embryonic Leg Muscle Development Differences between Wuzong and Shitou Geese

**DOI:** 10.3390/ijms24087188

**Published:** 2023-04-13

**Authors:** Xumeng Zhang, Yong Li, Chenyu Zhu, Fada Li, Zhiyuan Liu, Xiujin Li, Xu Shen, Zhongping Wu, Mengsi Fu, Danning Xu, Yunbo Tian, Yunmao Huang

**Affiliations:** College of Animal Science & Technology, Zhongkai University of Agriculture and Engineering, Guangzhou 510225, China

**Keywords:** DNA methylation, WGBS, goose, embryonic myogenesis, *MyoD1*

## Abstract

Skeletal muscle development from embryonic stages to hatching is critical for poultry muscle growth, during which DNA methylation plays a vital role. However, it is not yet clear how DNA methylation affects early embryonic muscle development between goose breeds of different body size. In this study, whole genome bisulfite sequencing (WGBS) was conducted on leg muscle tissue from Wuzong (WZE) and Shitou (STE) geese on embryonic day 15 (E15), E23, and post-hatch day 1. It was found that at E23, the embryonic leg muscle development of STE was more intense than that of WZE. A negative correlation was found between gene expression and DNA methylation around transcription start sites (TSSs), while a positive correlation was observed in the gene body near TTSs. It was also possible that earlier demethylation of myogenic genes around TSSs contributes to their earlier expression in WZE. Using pyrosequencing to analyze DNA methylation patterns of promoter regions, we also found that earlier demethylation of the *MyoD1* promoter in WZE contributed to its earlier expression. This study reveals that DNA demethylation of myogenic genes may contribute to embryonic leg muscle development differences between Wuzong and Shitou geese.

## 1. Introduction

Shitou (STE) and Wuzong (WZE) geese are vital indigenous goose breeds in Guangdong province, which is the largest goose producer and consumer in China. STE are mainly raised in the eastern coastal area of Guangdong province, and WZE are mainly raised in the northeastern area of the Pearl River Delta in Guangdong province. With an average weight of 8–13 kg at 70–90 days of age, STE are one of the largest goose breeds in the world. WZE are among the smallest goose breeds in the world, normally weighing 2–3 kg at 70–90 days of age [1]. Although the two goose breeds living in Guangdong province have similar growth cycles, their growth performance, as measured by growth rate and body weight, is quite different. According to our previous study, STE showed more intense pectoral muscle development than WZE from E15 to the day of birth [2]. Even so, whether the same developmental pattern applies to leg muscles and the molecular mechanisms regulating skeletal muscle development in the two goose breeds remains unclear.

Epigenetic mechanisms are hereditary modifications that regulate gene transcription without altering the DNA nucleotide sequence [3]. Various biological processes are affected by DNA methylation, including cell differentiation, tissue-specific gene expression, imprinting, and diseases [4]. DNA methylation occurs primarily at C nucleotides in vertebrate CpG dyads, and DNA methyltransferases (DNMTs) catalyze this process. Among them, *Dnmt1* primarily plays a role in maintaining DNA methylation patterns during replication. *Dnmt3a* and *Dnmt3b*, on the other hand, are enzymes that build DNA methylation patterns de novo. Each of these three DNMTs is involved in regulating development, and deficiency in any of them causes either embryonic lethality in mice or pups to die shortly after birth [5].

Through oxidation of 5mC to 5-hydroxymethylcytosine (5hmC), 5-formylcytosine (5fC) and 5-carboxylcytosine (5caC), followed by replication-dependent dilution or thymine DNA glycosylase (TDG)-dependent base excision repair, DNA methylation in the form of 5-methylcytosine (5mC) can be actively reversed to unmodified cytosine (C) [6]. DNA demethylation and oxidized 5mC play a role in preimplantation embryo development, primordial germ cell development, pluripotency, and differentiation [7]. TET methylcytosine dioxygenases (TET1, TET2, TET3) actively demethylate 5-methylcytosine (5mC) and produce hypomethylation at key regulatory regions [8].

In general, animals with a greater number of myofibers achieve higher growth rates [9]. The number of myofibers, which is normally fixed around hatching in poultry, is closely related to muscle mass [10]. Therefore, skeletal muscle development during embryonic stages to hatching is critical for muscle growth. In a previous study, the histological morphology of the pectoral and leg muscles of Zhedong White Goose was observed on embryonic day 7 (E7), E11, E15, E19, E23, and E27. The results showed that a large number of myoblasts appeared in Zhedong White Goose on E7–E11. During E11–E15, the number of myoblasts began to decrease, and differentiation occurred. A large number of myotubes appeared on E19, and myofibers began to appear on E23, and leg muscle development occurred significantly earlier than that of pectoral muscle [11].

In poultry, embryonic skeletal muscle development is critical for muscle growth and is precisely controlled by a series of genes expressed at different developmental stages. Myogenic regulatory factors (MRFs) are encoded by a gene family that is responsible for myogenesis, containing four members: *MyoD1*, *Myf5*, *Myogenin*, and *Myf6*. MRFs have a conserved basic helix-loop-helix (bHLH) domain that facilitates myogenesis [12]. In particular, *Myf5* and *MyoD1* initiate myogenesis, while *Myogenin* and *Myf6* mainly regulate myoblast differentiation [12]. These myogenic factors allow muscle cells to function properly by regulating myogenic progression and the expression of important genes for muscle cell function, such as myocyte enhancer factors (MEFs), *Igf1*, and myosin heavy chain (MyHC) [13,14].

Reprogramming of DNA methylation during embryonic development has been observed in human and mouse embryos. After fertilization, DNA methylation is largely erased, and after implantation, the DNA methylation level increases [15]. After implantation, the genes that gained DNA methylation are predominantly involved in early developmental processes, while the genes that lost DNA methylation are predominantly involved in tissue-specific developmental processes [16].

In our previous study of DNA methylation patterns during longissimus dorsi muscle development in embryos between a Wuzhishan pig (WZS, miniature pig) and a Landrace pig (LR, large-sized pig), we found that precocious myoblast differentiation in miniature pigs is due to earlier demethylation of myogenic genes [17]. Recent studies have investigated DNA methylation in multiple tissues in several poultry breeds. A comparison of methylome profiles among layer and broiler chicken lines suggested that low methylation in broilers might contribute to muscle development in the embryonic period [18]. A significant correlation was found between the DNA methylation levels of CpG sites of the *TNNI1* promoter and mRNA expression in the embryonic skeletal muscles of Gaoyou ducks [19]. It was found that DNA methylation affects the expression of muscle-related genes by modulating the accessibility for myogenesis transcription factors, indicating the involvement of the DNA methylation/*SP1*/*IGF2BP3* axis during porcine skeletal muscle development [20]. The process of DNA methylation is important for many biological processes; however, it is not yet clear how early embryonic muscle development differs between goose breeds of different body size.

In this study, we performed phenotype analysis and whole-genome bisulfite sequencing (WGBS) on leg muscle tissues of WZE and STE geese during early embryonic development. An overview of DNA methylomes was compiled, and their relationship to gene expression was determined. Several candidate differentially methylated genes (DMGs) that may be involved in the regulation of skeletal muscle development during early embryonic development in geese have been identified.

## 2. Results

### 2.1. Embryonic Leg Muscle Development at E23 Was More Intense in STE Than in WZE

In order to examine differences in leg muscle development between STE and WZE geese during embryonic stages, leg muscle development was assessed at E15, E23, and P1 in both breeds with H&E staining.

Myofiber density increased sharply during E15–E23 and decreased slightly during E23–P1 in both breeds. Meanwhile, myofiber diameter showed an increasing trend during E15–P1 in both breeds. It was found that myofiber density was significantly higher, and the myofiber diameter was significantly smaller in STE than in WZE at E23. Meanwhile, at E15 and P1, there appeared to be no difference in myofiber diameter and density between the two goose breeds (Figure 1A–C). These results indicate that embryonic leg muscle development at E23 was more intense in STE than in WZE.

### 2.2. WZE Showed Faster DNA Demethylation than STE during E23-P1

We obtained 2,387,710,186 mapped reads from the WGBS data for STE and 2,262,714,920 for WZE with mapping ratios of 81.96% and 81.68%, respectively (Appendix A). Throughout the genome, gene regulatory regions, and transcriptional elements, CG methylation was the strongest methylation form in all samples (64.96% on average), while CHH and CHG methylation rates ranged between 0.69% and 0.88% (Appendix A). Therefore, we focused on CG methylation patterns in this study.

Principal component analysis (PCA) analysis showed that replicates of each stage from both breeds had good consistency. Further, the CG methylation patterns of WZE and STE can be clearly separated into two clusters, and in both breeds, the CG methylation patterns of E15 were separated from those at E23 and P1 (Figure 2A).

The CG methylation level of STE showed no significant changes during E15–P1 in all genomic and gene regulatory regions we analyzed, while the CG methylation level of the genome, upstream 2 kb, downstream 2 kb, and 5′-UTR decreased significantly during E15–P1 in WZE (Figure 2B). Moreover, the CG methylation levels of STE were all higher than those of WZE in all genome and gene regulatory regions analyzed, among which CG methylation levels of the genome, gene body, upstream 2 kb, downstream 2 kb, exon, coding sequence, and intron reached a significant level (Figure 2B,C). Gene expression is mainly regulated by the region around the TSS. We observed a dramatic decrease in DNA methylation in the region upstream of the TSS, a sharp increase in the regions surrounding the gene body, and a plateau until the TTS (Figure 2C). While DNA methylation level of CpG islands, shores and shelves showed no significant difference between breeds at same stage (Appendix A).

These findings indicate that DNA methylation patterns vary between breeds and stages, especially during E23–P1, with WZE showing faster DNA demethylation processes than STE.

### 2.3. WGBS-Transcriptomic Data Integration

For more insight into how DNA methylation is correlated with gene expression, we examined DNA methylation levels (CG) in gene regulatory regions (upstream 2 kb, gene body, and downstream 2 kb). We classified genes into four categories based on RNA-seq data (high expression, middle expression, low expression, and no expression). In upstream 2 kb (especially regions near TSSs) and downstream 2 kb (except in the no expression group), a negative correlation between gene expression and DNA methylation was found in all three stages (Figure 3), which means higher gene expression in this region is associated with lower DNA methylation levels. Furthermore, a positive correlation was found in the gene body region near TTSs in all three stages (except the high expression group) (Figure 3). In summary, gene expression and DNA methylation levels of different gene regions were not always similarly correlated in both goose breeds.

### 2.4. Analysis of DEGs during Embryonic Development in Relation to DNA Methylation

To further identify genes regulated by DNA methylation during goose embryonic development, we conducted an integrated study of DMR-related genes and DEGs in gene regulatory regions (upstream 2 kb, gene body, and downstream 2 kb) at the same stage between the two goose breeds.

The number of DMR-related genes increased, while the number of DEGs showed a slight change during E15–P1. In total, 38, 31, and 35 genes that were both DMR-related genes and DEGs were identified at E15, E23, and P1 in the comparison between the two goose breeds, respectively (Figure 4A). At E15, the most genes were genes with upregulated expression and upregulated DNA methylation (E+&M+), followed by upregulated expression and downregulated DNA methylation (E+&M−) and downregulated expression and upregulated DNA methylation (E−&M+). At E23 and P1, the most genes were E−&M+, followed by E+&M+ and E−&M− (Figure 4A).

Heatmaps of gene expression patterns were analyzed based on 104 genes that were both DMR-related genes and DEGs (Figure 4B). Most genes were gene body DMR-related genes (90 genes), followed by downstream 2 kb (11 genes) and upstream 2 kb (3 genes) (Figure 4B). Several gene expression clusters could be found in the heatmap, and most of the genes showed higher expression in STE than in WZE during all three stages. These results again indicate that DEGs associated with DNA methylation showed different expression and DNA methylation patterns at E15 between two goose breeds, and the different expression patterns of these DMR-related DEGs may play a vital role in the distinct regulation of myogenic processes between these two goose breeds.

### 2.5. GO Analysis of Methylation-Related DEGs

To understand the biological functions of methylation-related DEGs, GO analysis was conducted on all 104 DEGs between the two goose breeds at the same stage. Four significantly enriched myogenesis-related GO terms were identified in the biological process category: myosin filament organization, cell differentiation, myosin II complex, and myosin filament (Figure 5). Among these biological processes, myosin filament organization was significantly enriched in WZE E15 vs. STE E15 (CG, all gene, and gene body), WZE E23 vs. STE E23 (CG, all gene, and gene body); cell differentiation was significantly enriched in WZE E23 vs. STE E23 (CG, all gene, and gene body); myosin II complex was significantly enriched in WZE E23 vs. STE E23 (CG, all gene, and downstream 2 kb); and myosin filament was significantly enriched in WZE E15 vs. STE E15 (CG, all gene, and gene body). These findings indicate that DNA methylation had a more distinct impact on gene regulation during E15–E23 between two goose breeds. Several genes involved in the abovementioned myogenesis-related biological processes were identified, such as *Obscn*, *Myo5B*, *NRAP*, *Myh1B*, and *KIF24* (Appendix A). These genes may be affected by different DNA methylation patterns to regulate embryonic leg muscle development in the two goose breeds.

### 2.6. Myogenic Genes May Be Expressed Earlier in WZE as a Result of Earlier Demethylation around TSS Region

Furthermore, in order to determine whether DNA methylation affects goose embryonic myogenesis by affecting the expression of myogenic genes, several well-known genes involved in DNA methylation, muscle fate determination, proliferation, differentiation, and genes identified from GO analysis of methylation-related DEGs were selected for gene expression and DNA methylation analysis.

All DNA methylases/demethylases showed decreased expression during E15–P1. Among the three methylase-encoding genes, *Dnmt1* and *Dnmt3B* showed significantly higher expression in STE than in WZE at E15, while *Dnmt3A* and *Dnmt3B* showed significantly higher expression in WZE than in STE at E23. Among the three demethylase-encoding genes, *Tet1* showed significantly higher expression in WZE than in STE at E15, while *Tet2* and *Tet3* showed significantly higher expression in WZE than in STE at E23. At P1, *Tet2* showed significantly higher expression in STE than in WZE. Among myogenic genes, *MyoD1* and *Obscn* showed significantly higher expression in WZE than in STE at E15, while *Igf1* showed significantly higher expression in WZE than in STE at E23 (Figure 6A and Appendix A).

Average DNA methylation levels around TSS regions and gene bodies of several myogenic genes from WZE and STE during E15–P1 were analyzed and visualized by IGV software (Figure 6B and Appendix A). It was found that *MyoD1*, *Myogenin*, *Igf1*, *Obscn*, *Kif24*, *Myo5B*, *Myh1b*, *Myh7*, and *Igf1R* all showed lower DNA methylation level around TSS region in WZE than in STE at least in one stage (E15, E23, or P1).

These results indicate that DNA methylation may affect the expression of goose myogenic genes and that myogenic genes in WZE may undergo DNA demethylation and methylation reconstruction earlier than in STE.

### 2.7. Earlier Demethylation of the MyoD1 Promoter in WZE May Lead to Its Earlier Expression

First, RNA-seq results of four myogenesis-related DEGs related to DNA methylation or vital myogenic genes were validated by RT-qPCR, using *Gapdh* as an endogenous control. RT-qPCR and RNA-seq correlation coefficients were computed by SPSS, and the average r value was over 0.92 (Figure 7A and Appendix A). Based on these results, the two methods showed a high level of consistency. The results again proved that *MyoD1* showed significantly higher expression in WZE than in STE at E15.

As *MyoD1* plays a decisive role in embryonic myogenesis, DNA methylation patterns of the promoter and first exon regions at each stage from both goose breeds were visualized by IGV software to analyze its correlation with *MyoD1* expression. It was found that STE showed a higher DNA methlylation level than WZE in the promoter and first exon regions of *MyoD1* during all three stages (Figure 7B and Appendix A). These results were in line with our previous study, which showed that DNA methylation in the promoter and first exon regions of *MyoD1* and gene expression were negatively correlated [17].

Furthermore, to validate WGBS data, pyrosequencing was conducted to analyze the DNA methylation levels of the *MyoD1* promoter region. By pyrosequencing, the methylation status of the *MyoD1* promoter (−480 bp to −723 bp), which contains 13 CG sites, was studied during E15–P1 (Figure 7C). WZE showed significantly lower DNA methylation levels in three CG cites, and most CG sites in the *MyoD1* promoter showed lower methylation levels in WZE than in STE at E15 and E23, which was in agreement with the WGBS results. At P1, two CG sites showed significantly lower DNA methylation levels and three CG sites showed significantly higher DNA methylation levels in the *MyoD1* promoter in WZE than in STE (Figure 7D and Appendix A). These results indicate high consistency between WGBS and pyrosequencing data, and earlier demethylation of the *MyoD1* promoter in WZE may lead to its earlier expression.

## 3. Discussion

Our study reveals that the embryonic leg muscle development at E23 is more intense in STE than in WZE. In our previous study concerning pectoral muscles of STE and WZE, we found that STE had a significantly higher myofiber density during E15–P1 and a significantly larger myofiber diameter at E15 than WZE [2], indicating that embryonic leg and pectoral muscle development of WZE and STE showed different patterns. In our previous studies concerning pigs of different body sizes, as well as chicken and quail, it was found that embryonic muscle development was slower in breeds/species with a larger body size, which was inconsistent with the observations in geese [17,21]. Therefore, embryonic muscle development patterns of goose breeds differing in body size may be different from those in the aforementioned species.

In this study, we found a negative correlation between muscle phenotype and DNA methylation. A similar correlation was found in the pig longissimus dorsi muscle, where a global loss of DNA methylation in the gene body region in skeletal muscle of middle-aged pigs was found compared with the young group [22]. Similar to this study, species-specific or breed-specific DNA methylation patterns were also observed when comparing human, bovine, and mouse DNA methylation patterns [23]. In this study, we found that gene expression and DNA methylation were negatively correlated only in regulatory regions around TSSs. A positive correlation was found between gene expression and DNA methylation in gene bodies near TTSs, which is in agreement with our previous study in pigs [17] as well as previous studies in humans and cattle [24,25].

In our study, it was found that most of the genes showed higher expression in STE than in WZE. Meanwhile, DEGs associated with DNA methylation showed different expression and DNA methylation patterns in the early embryonic stage (E15) between two goose breeds. In our previous study, we also found that E15 is crucial for distinct embryonic pectoral muscle development characteristics between WZE and STE [2]. It was reported that *CnAα*, *NFATc3*, and *IGF-1* mRNA, which is involved in the regulation of myocyte hypertrophy and fiber type specificity, were detected as early as embryonic day 13, and the highest level was also found at this stage in both pectoral and leg muscle of Gaoyou and Jinding ducks [26]. A recent study investigated known key regulatory genes during pectoral and leg muscle development in Cornish, White Plymouth Rock, White Leghorn, and Beijing-You Chickens, and found that the highest expression level of *Myomaker* occurred from embryonic days E13 to E15 for all breeds, indicating that it was the crucial stage of myoblast fusion [27]. These findings indicate that E15 or earlier embryonic stages may also be crucial stages regulating embryonic skeletal muscle development in geese, which needs further investigation.

In this study, during E15–E23, DNA methylation had a more distinct effect on gene regulation between the two goose breeds. *Obscn*, *Myo5B*, *NRAP*, *Myh1B*, and *KIF24* were identified as genes involved in myogenesis-related GO terms in the biological process category whose expression is affected by DNA methylation. *Obscn* was originally discovered in striated muscles as a cytoskeletal protein with scaffolding and regulatory functions [28]. *Myo5B* was found as a differentially expressed gene in the longissimus and leg muscle of rabbit breeds [29]. *NRAP* is primarily associated with developing myofibrillar structures containing alpha-actinin, but not mature myofibrils [30], and it was found that *NRAP* expression during prenatal development of skeletal muscle differed by more than two-fold between Pietrain and Duroc pigs, whose muscularity and muscle structure are markedly different [31]. In response to intensive genetic selection, genomic regions harboring genes encoding MYH proteins have been identified, including *Myh1B* [32], and LncRNA-FKBP1C was found to bind with *MYH1B* and enhance its protein stability, affecting proliferation, differentiation, and fiber type conversion in skeletal muscle cells [33]. The expression of these genes and myogenic genes identified in this study may be affected by different DNA methylation patterns to regulate embryonic leg muscle development in the two goose breeds.

In this study, it was found that DNA methylation may affect the expression of goose myogenic genes. Meanwhile, myogenic genes in WZE showed earlier trends of DNA demethylation and methylation reconstruction processes than in STE, which is in agreement with our previous study in pigs [17]. In addition, we examined the methylation and transcription regulation of vital myogenic cell fate determining gene-*MyoD1*, which can transition the lineage of multiple non-muscle cells into muscle cells [34]. Our study indicates that earlier demethylation of the *MyoD1* promoter in WZE may lead to its earlier expression.

Some studies have shown that the 20 kb core enhancer upstream of the *MyoD1* promoter plays a key role in the appropriate spatiotemporal expression of *MyoD1* in mouse embryos [35]. Knockout of this enhancer in mouse embryos resulted in a *MyoD1* expression delay of 1–2 days, thus delaying myogenesis by 1–2 days [36]. In a mouse embryo and muscle cell lines, the core enhancer of *MyoD1* is almost completely demethylated, while in non-muscle cell lines and non-muscle tissues, the methylation degree of enhancer is relatively high [37]. Although *MyoD1* have long been reported to play a decisive role in the process of muscle formation, the specific regulatory mechanisms of *MyoD1* in the process of goose muscle formation, especially its role in regulating goose with different body size, are still unclear.

Direct evidence concerning the role of DNA methylation in the observed phenotypic differences between the two breads was not studied, and some relationship between results were negative or could not be solved in this study. Therefore, in future studies, we need to further investigate the role of Dnmts and Tets in goose embryonic muscle development and determine the DNA methylation differences of gene regulatory elements (such as enhancers) of vital myogenic genes in vivo and vitro between two goose breeds.

## 4. Materials and Methods

### 4.1. Animals and Sample Collection

STE and WZE geese in this study were purchased from Guangdong Qingyuan Jinyufeng Goose Co., Ltd. Goose embryos/goslings with similar embryonic development status and weight were selected for different breeds at each stage. The breeding geese had similar feeding conditions. The feeding and management conditions of breeding geese, such as feed and light, met the national standard criteria. All geese completed the immunization program and were in good health. Embryos were collected on E15 (*n* = 3) and E23 (*n* = 3) after fertilization, while goslings were slaughtered on post-hatch day 1 (P1, *n* = 3) for each breed. Leg muscle samples were collected and snap-frozen in liquid nitrogen or fixed in 4% paraformaldehyde solution for hematoxylin and eosin (H&E) staining.

### 4.2. H&E Staining

Leg muscle samples were first embedded in paraffin. Sections were then cut at the same position and H&E staining was conducted as previously described [38]. Micrographs were taken with an Axio Imager Z1 (Zeiss) for each area (400×). The number/diameters of myofibers of one randomly selected part were counted/measured in at least six slides by CaseViewer 2.4.0. At least three visual fields of each slice were randomly selected for photographing. The nucleus is blue, and the cytoplasm is red. Image Pro Plus 6.0 was used to count the cross-sectional images of myofibers. More than six myofibers or muscle bundles were randomly selected from each image, the diameter of myofibers (mm) or the cross-sectional area of muscle bundles (mm^2^) was measured, and the average value was calculated. The number of myofibers or muscle bundles in each visual field was counted, the total cross-sectional area of myofibers (mm^2^) was measured, and the density of myofibers or muscle bundles per unit area (n/mm^2^) was calculated.

### 4.3. Sequencing and Analysis of Methylomes

DNA extraction: A summary of the study design and data analysis procedure can be found in Appendix A. Using leg muscle tissue samples from each breed collected at three developmental stages mentioned above, we extracted and sequenced DNA independently and created 18 DNA libraries. The DNA samples were dissolved in 200 µL of lysis buffer from the DNA Micro Kit (Catalog no. 56304, Qiagen, Germany) and then incubated with proteinase K for 36 h at 56 °C. We extracted DNA using the QIAamp DNA Micro Kit (Qiagen) and determined its concentration at 260 nm using a NanoDrop ND-1000 spectrophotometer (Nanodrop Technologies Inc., Charlotte, NC, USA).

DNA conversion and sequencing: Each sample was bisulfite-modified with the EZ DNA methylation kit (Zymo Research, Orange, CA, USA) using 500 ng of DNA. Sequencing and data processing were carried out as described previously [39]. The DNA was fragmented using sonication into 100–500 bp fragments, followed by blunting, dA addition at the 3′end, and adapter ligation. In order to monitor the bisulfite conversion efficiency, the adapter sequence contained multiple methylcytosines. An alternative protocol was used to convert unmethylated cytosines to uracils following bisulfite treatment [40]. Gel-purified DNA fragments with sizes ranging from 320 to 380 bp were sequenced according to the manufacturer’s instructions. DNA that had been converted was sequenced by Illumina Solexa GA on an Illumina system (Illumina, San Diego, CA, USA) using 50-bp paired ends. Raw data were processed using the Illumina Pipeline v1.3.1.

WGBS mapping and initial reads processing: Using SOAP aligner v2.21 http://soap.genomics.org.cn (accessed on 11 March 2022), short reads generated by Illumina sequencing were aligned to the *Anser cygnoides* reference genome assembly AnsCyg_PRJNA183603_v1.0, ftp://ftp.ncbi.nlm.nih.gov/genomes/refseq/vertebrate_other/Anser_cygnoides/latest_assembly_versions/GCF_000971095.1_AnsCyg_PRJNA183603_v1.0 (accessed on 23 November 2021) [41]. *Anser cygnoides* has strand-specific DNA methylation, so its plus strand and minus strand should be separated and formed into alignment target sequences. To obtain high quality clean reads, raw reads were filtered according to the following rules: (1) removing reads containing more than 10% of unknown nucleotides (N); (2) removing low quality reads containing more than 40% of low quality (Q-value ≤ 20) bases. BSMAP software (version 2.90) was used for alignment, and default parameters were used. The reads that mapped to the same start position were considered clonal duplications generated during the PCR process, and only one of them was kept. To detect mCs, we transformed each aligned read and both strands of the *A. cygnoides* genome back into their original forms and then mapped them together. An mC is defined as a cytosine in the WGBS read that is also matched to the same cytosine in the plus strand, or as a guanine in the minus strand in the absence of matching cytosines. The Q score, which is used in base-calling pipelines (SolexaPipeline-1.0) (Illumina) to detect sequences from fluorescent images, is calculated as follows:$$Q=10\log_{10}[p\left(X\right)/\left(1-p\left(X\right)\right)]$$
where *p*(*X*) is the probability of correctly calling a read. We then filtered out all potential mCs with Q scores of less than 20, guaranteeing that a base is correctly called in more than 99% of cases, based on highly conservative assumptions [42].

Identification of mCs: The first step was to check the mCs detected by the WGBS of each library. In a read where mC at a non-CG site actually becomes G, it is likely to be a false-positive non-CG methylation, although it may be a CG methylation due to polymorphism between individuals. Nevertheless, in order to be conservative, we always excluded these potential mCs caused by single-nucleotide polymorphisms. As a background noise control, we used the methylation rate at non-CG sites of the whole genome, which is a measure of the false-positive rate (errors resulting from thymidine-to-cytosine conversion and non-conversion rate) [42]:$$\mathrm{false}\text{-}\mathrm{positive}\;\mathrm{rate}=\left(n_{\mathrm{mCHH}}+n_{\mathrm{mCHG}}\right)/n_{\mathrm{depth}}\times 100\%.$$

The distribution of methylated C bases on the genome includes three forms (CG, CHG, and CHH, where H represents A, T, or C). n_mCHG_ and n_mCHH_ are the total number of sequenced Cs in CHG and CHH contexts in the reference genome, respectively. The n_depth_ value represents the total sequencing depth at cytosine positions in CHG and CHH contexts. As a measure of the false mC discovery rate, we set a significance threshold (99% confidence) to identify the presence of an mC at each base position based on binomial probability distribution, the read depth, and the false-positive rate. Sites with mCs below the minimum threshold were rejected. The minimal number of reads for a CG to be included in the study is 4. The definition of CpG island is CG content > 50%, length > 200 bp, observed CpG/expected CpG > 0.6. CpG shores refers to the area located within 2 kb upstream and downstream of CpG island; CpG shelves refers to the area located within 2 kb upstream and downstream of CpG shores.

### 4.4. Sequencing and Processing of Transcriptome Data

Using the same samples as for WGBS, we constructed RNA libraries and performed deep sequencing. In order to build and sequence mRNA libraries, total RNA was extracted from frozen muscle tissues using TRIzol reagent (Invitrogen, Carlsbad, CA, USA) following the manufacturer’s protocols. RNA integrity and concentration were assessed using an Agilent 2100 Bioanalyzer (Agilent Technologies, Palo Alto, CA, USA). We prepared sequence tags using Illumina’s Digital Gene Expression Tag Profiling Kit according to the manufacturer’s instructions. RNA libraries were constructed, and sequencing was performed by the Beijing Genomics Institution (BGI) on an Illumina Genome Analyzer using RNA integrity number (RIN) ≥ 7.0 and 28S/18S ≥ 1.0.

By filtering raw data, clean tags were obtained by removing adaptor tags, low quality tags, and tags with a copy number equal to 1. The clean tags were classified according to their copy numbers in the library. The proportion of each category was calculated in relation to the total clean tags. Clean distinct tags were analyzed similarly. The sequencing library was analyzed by BGI for saturation. Fragments per kilobase of exon model per million mapped fragments (FPKM) values were used to normalize gene expression levels. Sequences were obtained from *A. cygnoides* RefSeq databases [43]. The consistency of cluster analysis of RNA-seq is good among different replicate samples from the same breeds at the same time points (Appendix A).

### 4.5. Bioinformatic Analyses

We constructed global DNA methylation profiles by dividing the *A. cygnoides* genome into 10 kb windows. We then visualized the global patterns using Integrative Genomics Viewer (IGV) software [44]. Using Ward’s minimum variance method implemented in methylKit, we computed clusters based on methylation levels and patterns of different samples [45]. Annotation information for the goose reference genome was obtained from public FTP site ftp://ftp.ncbi.nlm.nih.gov/genomes/refseq/vertebrate_other/Anser_cygnoides/latest_assembly_versions/GCF_000971095.1_AnsCyg_PRJNA183603_v1.0 (accessed on 23 November 2021). We considered the genomic region from the transcription start site (TSS) to the transcription termination site (TTS) as the gene body region, and we considered the 2 kb genomic region upstream of the TSS as the proximal promoter region. Quantile regression was used to calculate *p* (R package quantreg http://cran.r-project.org/package=quantreg, accessed on 21 March 2022). Figures were created with ggplot2 [46]. An online Venn diagram tool was used to create Venn diagrams (http://bioinformatics.psb.ugent.be/webtools/Venn/, accessed on 2 May 2022). BMK cloud heatmaps were generated by BioCloud (http://www.biocloud.net, accessed on 18 May 2022).

DMR identification: Based on the algorithm described previously, the CpGs showing gains or losses in methylation were identified [47]. Differentially methylated regions (DMRs) were defined by three consecutive CpGs that matched a criterion (that is, gain/loss of methylation, with a Benjamini–Hochberg corrected *p* < 0.01, FDR < 0.01) and had a maximum of 1 kb between CpGs and the highest three CpGs that did not match the criterion. The identification requirements of DMR are as follows: for the CG difference in a certain range, we require that the number of CG in the window should not be less than 5, the absolute value of methylation difference should be ≥0.25, and the Q value should be ≤0.05. A DMG is a gene that has either gained or lost methylation (|log2(fold change [FC])| ≥ 1) and has a false discovery rate (FDR) of at least 0.001 (adjusted by the Benjamini–Hochberg method).

Gene expression data: For FPKM expression levels, transcriptional data from 18 libraries were used, and edgeR 3.2.4 was used to calculate the FPKM expression levels [48]. Across multiple libraries, genes were considered as differentially expressed (DEGs) if |log2(FC)| ≥ 1 and FDR ≤ 0.05. FPKM ≥ 100 was regarded as high expression; 10 ≤ FPKM < 100 was regarded as middle expression; 0 ≤ FPKM < 10 was regarded as low expression; FPKM = 0 was regarded as no expression.

Gene ontology (GO) and KEGG pathway analyses: By using DAVID, the methylated CpGs were mapped to their nearest gene (that is, to the nearest TTS of a gene) and tested for enrichment of GO terms [49]. An FDR of 0.05 was used as the cut-off for enrichment in GO terms for methylated CpGs. A background set containing the nearest genes of all CpGs covered on the reference genome was used.

Analyses of methylation distributions on average: We determined the distribution of average CG methylation levels in the normalized models of the 2 kb, 5 kb, or 10 kb regions upstream of the TSS, gene body (first exon, first intron, internal exon, internal intron, and last exon), and the 2 kb, 5 kb, or 10 kb regions downstream of the TTS in each stage. Dots indicate the mean methylation level per bin, and lines represent the 5-bin moving average. A 2 kb, 5 kb, or 10 kb region upstream and downstream of each gene was divided into 100 bp intervals. The intervals for each gene were 20, 50, or 100 (5% per interval).

### 4.6. Real-Time Quantitative PCR

Real-time quantitative PCR (RT-qPCR) was performed on total RNA used in RNA-seq to validate gene expression and investigate the relationship between gene expression and DNA methylation. The PrimeScript^TM^ RT Master Mix kit (Takara, Dalian, China) was used to synthesize cDNA using oligo (dT) and random hexamer primers. RT-qPCR was conducted using a standard SYBR^®^ Premix Ex Taq^TM^ II (Takara, Dalian, China) on a Bio-Rad CFX96 Real-Time PCR Detection System (Bio-Rad, Hercules, CA, USA) in accordance with the manufacturer’s instructions. The primer sequences used for RT-qPCR are listed in Appendix A. Each RNA sample was analyzed in triplicate. The data were normalized to the geometric mean of the data from the goose *Gapdh* gene. We calculated the relative expression levels of the target mRNAs using the 2^−ΔΔCt^ method.

### 4.7. MyoD1 CpG Island Pyrosequencing

We used genomic DNA from the same WZE and STE samples as for WGBS. Following extraction, 1 μg of genomic DNA was converted using the EpiTect Bisulfite Kit (QIAGEN, Shanghai, China) according to the manufacturer’s instructions. Unmethylated cytosines were converted into uracils and methylated cytosines were protected after chemical conversion. The methylation status of the *MyoD1* gene promoter (−480 bp to −723 bp), which contains 13 CG cites, was analyzed by pyrosequencing. The details of primer design can be found in Appendix A. Pyrosequencing amplification and sequencing primers were designed using Assay Design Software in Pyrosequencing™ Systems (Biotage, Stockholm, Sweden). PCR analysis was conducted in triplicate for each sample. We amplified the promoter region of the goose *MyoD1* gene by nested PCR with two sets of primers. Sequencing was conducted using the Pyro Gold Reagent Kit (Biotage, Stockholm, Sweden) with a PSQ 96MA Pyrosequencing instrument (Biotage, Stockholm, Sweden).

## 5. Conclusions

In summary, this study showed that the embryonic leg muscle development at E23 is more intense in STE than in WZE, which may be due to the earlier demethylation of myogenic genes around TSSs. Additionally, demethylation of the *MyoD1* promoter in WZE may contribute to its earlier expression.

## Figures and Tables

**Figure 1 ijms-24-07188-f001:**
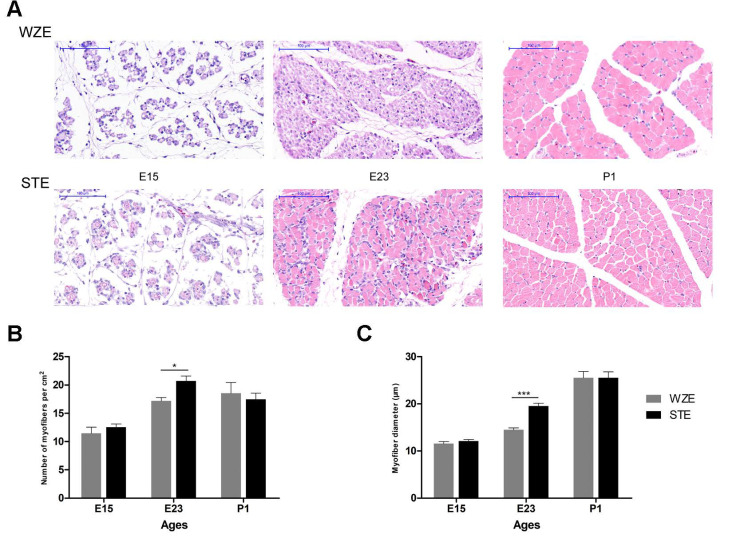
H&E staining results of leg muscle from STE and WZE at E15, E23, and P1. (**A**) H&E staining results of leg muscle from STE and WZE at E15, E23, and P1. (**B**) Statistical results of myofiber density (n/cm^2^). (**C**) Statistical results of myofiber diameter (μm). Scale bar = 100 μm. * *p* < 0.05; *** *p* < 0.001.

**Figure 2 ijms-24-07188-f002:**
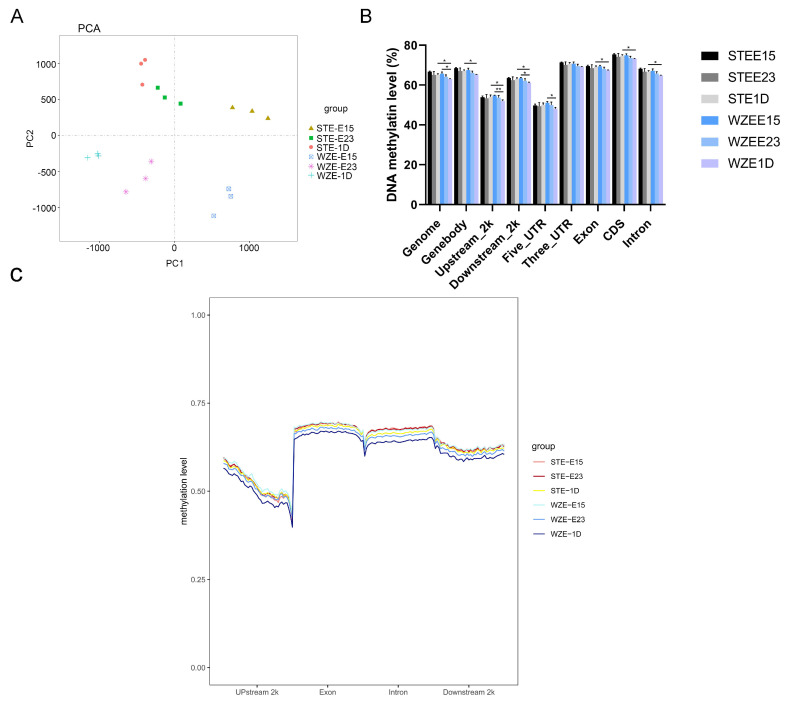
DNA methylome profiles of STE and WZE. (**A**) PCA of CG methylation patterns of STE and WZE during E15−P1. (**B**) CG methylation levels of both breeds in each stage in genome, gene body, upstream 2 kb, downstream 2 kb, 5′−UTR, 3′−UTR, exon, coding sequence, and intron regions of STE and WZE during E15−P1. (**C**) Average CG methylation level of all the libraries in upstream 2 kb, exon, intron, and downstream 2 kb regions of STE and WZE during E15−P1. * *p* < 0.05; ** *p* < 0.01.

**Figure 3 ijms-24-07188-f003:**
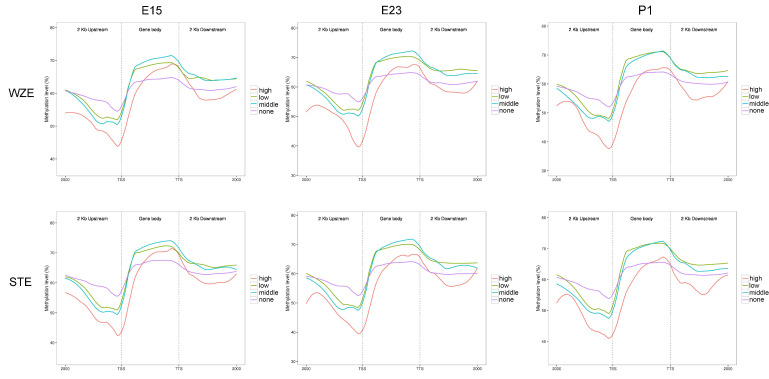
WGBS-transcriptomic data integration in gene regulatory regions between two goose breeds. According to RNA-seq data, genes were considered as differentially expressed (DEGs) if |log2(FC)| ≥ 1 and FDR ≤ 0.05. FPKM ≥ 100 was regarded as high expression; 10 ≤ FPKM < 100 was regarded as middle expression; 0 ≤ FPKM < 10 was regarded as low expression; FPKM = 0 was regarded as no expression. CG methylation levels in gene regulatory regions (upstream 2 kb, gene body, and downstream 2 kb) were analyzed in each group.

**Figure 4 ijms-24-07188-f004:**
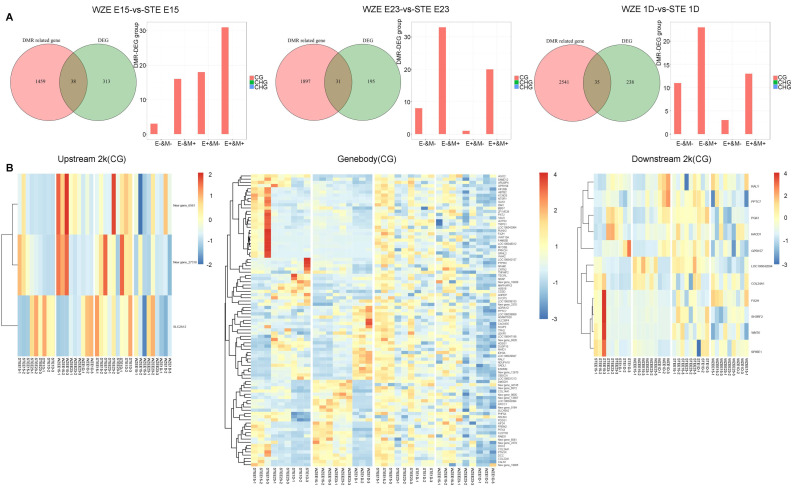
The identification of differentially expressed genes (DEGs) regulated by DNA methylation during embryonic development. (**A**) Venn diagram of DMR related genes and DEGs at the same stage between two goose breeds, together with statistical results of genes involved in different modes of DNA methylation and gene expression correlation. (**B**) Heatmap of gene expression patterns of genes that were both DMR−related genes and DEGs in upstream 2 kb, gene body, and downstream 2 kb regions (hierarchical cluster analysis on the left panel and non-hierarchical cluster analysis on the right panel). E, gene expression level; M, DNA methylation level; +, upregulation; −, downregulation.

**Figure 5 ijms-24-07188-f005:**
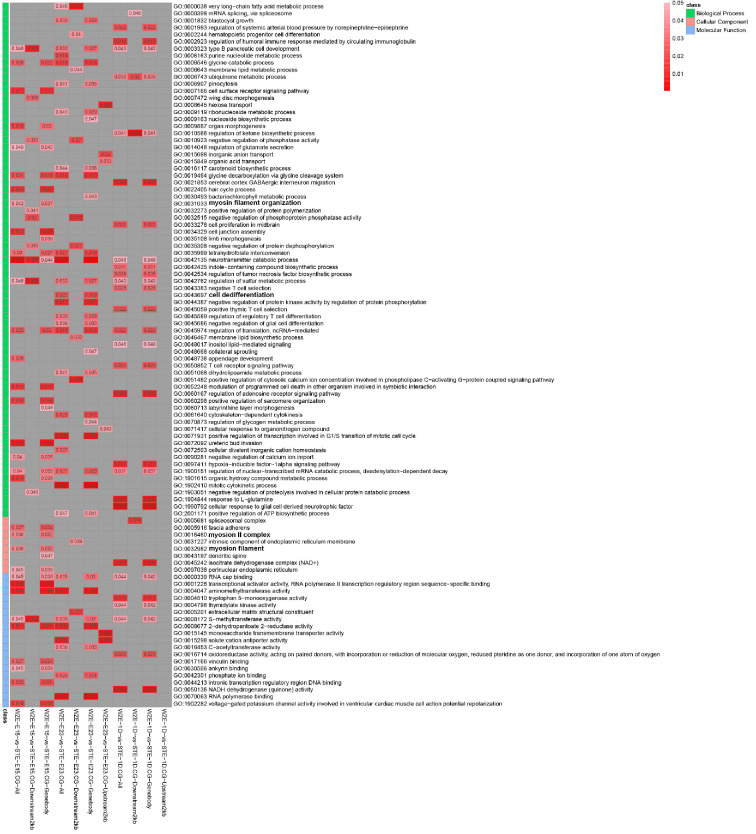
GO analysis of DNA methyation-related DEGs. Myogenesis-related biological processes are marked bold in the figure. Different colors on the left mean different GO classes; different colors and numbers in the figure indicate different corrected *p*-values; NA, not applicable.

**Figure 6 ijms-24-07188-f006:**
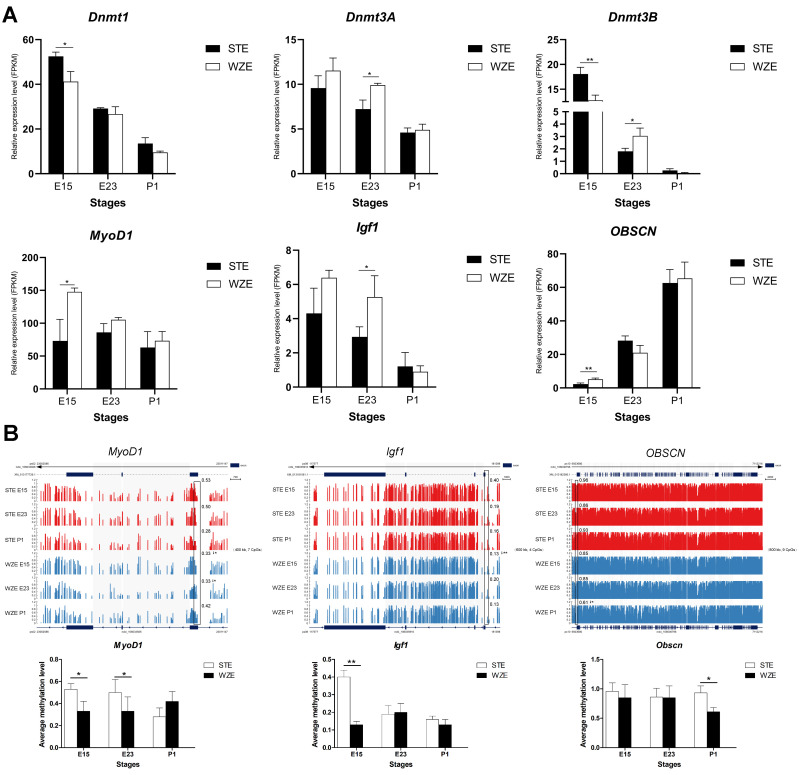
Gene expression patterns and DNA methylation level of vital DNA methylases and myogenic genes. (**A**) RNA-seq gene expression pattern of crucial DNA methylases and myogenic genes in STE and WZE during E15–P1 (*n* = 3). (**B**) DNA methylation level of myogenesis-related DEGs related to DNA methylation or vital myogenic genes in WGBS data illustrated by IGV software in STE and WZE during E15–P1. Regions around TSSs are marked in black boxes with average DNA methylation level shown beside the boxes. The length of and the number of CpGs in this box is marked in the figure. Quantitative analyses of indicated boxes are presented. Downward arrows indicate a lower DNA methylation level in comparison with STE. * *p* < 0.05; ** *p* < 0.01.

**Figure 7 ijms-24-07188-f007:**
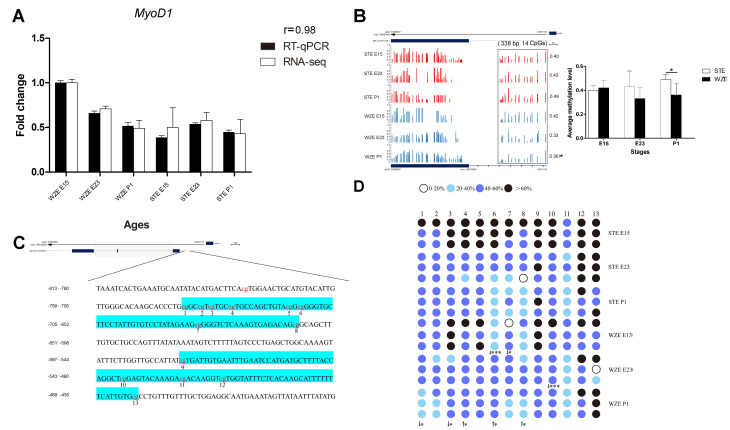
Validation of the association between DNA methylation and gene expression in the *MyoD1* promoter region. (**A**) RT−qPCR and RNA-seq validation of *MyoD1* expression (*n* = 3). (**B**) *MyoD1* promoter and first exon DNA methylation level as shown by IGV software based on WGBS data. Regions around TSSs are marked in black boxes with average DNA methylation level shown beside the boxes. The length and number of CpGs in this box is marked in the figure. Quantitative analyses of indicated boxes are presented. (**C**) A total of 13 methylation sites within the promoter region (−480 bp to −723 bp) of *MyoD1* were examined in WZE and STE by pyrosequencing. (**D**) Pyrosequencing results of 13 methylation sites within the promoter region of *MyoD1.* Different colors represent different DNA methylation levels, which are indicated in the figure. Downward arrows indicate a lower DNA methylation level in comparison with STE, upward arrows indicate a higher DNA methylation level in comparison with STE. * *p* < 0.05; *** *p* < 0.001.

## Data Availability

The raw RNA-seq data have been submitted to the SRA database under PRJNA665623, and the raw WGBS data have been submitted to the SRA database under PRJNA903391 (http://www.ncbi.nlm.nih.gov/Traces/sra/, accessed on 1 January 2023). Detailed WGBS and RNA-seq scripts are provided in the Appendix A.

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
