# Peer review of "DNA Demethylation of Myogenic Genes May Contribute to Embryonic Leg Muscle Development Differences between Wuzong and Shitou Geese"

_ijms, 2023, doi:10.3390/ijms24087188_

Round 1

Reviewer 1 Report

The manuscript submitted by Zhang et al investigates an interesting question, whether the difference between Wuzong and Shitu geese is due to DNA methylation difference between the two breads. They consider that yes, an important difference methylation difference in the developing embryonic leg contributes to quicker terminal differentiation. The reviewer is not convinced, please tone down the title.  

Comments:

1/ I don’t understand the role of the following sentence in the text. 

“The embryonic development of pineal metabolism begins with serotonin synthesis, followed by serotonin acetylation and 5-hydroxyindole methylation in goose (18).” 

Please remove.

2/ possitive ln 356 – please correct

3/The paragraph starting at ln 363 is a bit unclear: Please rephrase the paragraph and determine better the classes. : “There was a negative or positive correlation between DNA methylation and gene expression in certain regulatory regions in all stages we studied. According to RNA-seq data, genes were classified into four groups based on their FPKM expression levels (high expression, middle expression, low expression and no expression). CG methylation levels in gene regulatory regions (upstream 2 kb, gene body and downstream 2 kb) were analyzed in each group.“

4/ What was the minimal number of reads for a CG to be included in the study? Was there any methylation difference required for the identification of a DMR?

5/ Please check the methylation and expression level of the TeT genes. 

6/ Did you observe any methylation difference between Zwe and STE at enhancers, CpG shores and shelves?

7/ The quality of images is very low, sometimes it’s impossible to read the legends. On figure 5, the green/red P values scale is not explained. Are they corrected P values? The non-significant values should not be indicated with such a strong green color. I’d propose grey. The color-code might rather be changed to methylation difference of the significant DMRs. Please switch to blue-yellow coding.

8/ On figure 7b the rectangle really corresponds to 300 kb?

9/ Figure 7a is not really convincing. The MyoD1 expression kinetic trend determined by the two different methods is similar but quantitatively seems to be different, which might have physiologic relevance. Please repeat the qPCR experiment on new samples.

10/ DNA methylation level representation on 7d is not clear. Legend is missing. Please find a representation, which is more obvious. The greyscale doesn’t allow to easily identify the differences.

11/ The discussion is too lengthy. Please shorten. The reviewer considers that the presented results are rather negative, do not really show the role of the DNA methylation in the observed phenotypic difference between the two breads. Please include this in the discussion. Negative results are important!

Author Response

Dear Reviewer,

Thank you for your comments concerning our manuscript entitled "Earlier embryonic leg muscle myogenic gene demethylation contributes to more rapid myoblast terminal differentiation in Wuzong geese than in Shitou geese" (ijms-2266565). Those comments are all valuable and very helpful for revising and improving our paper, as well as the important guiding significance to our researches. We have studied all the comments carefully and modified the paper according to the reviewer’s suggestions. Revised portion are highlighted with tracked changes in the manuscript.

It would be greatly appreciated if our revised manuscript could be considered for publication in International Journal of Molecular Sciences.

Sincerely yours,

Xumeng Zhang

-----------------------------

The following is a point-to-point response to the reviewer’s comments.

Reviewer’s Comments to Author:

Reviewer 1:

The manuscript submitted by Zhang et al investigates an interesting question, whether the difference between Wuzong and Shitu geese is due to DNA methylation difference between the two breads. They consider that yes, an important difference methylation difference in the developing embryonic leg contributes to quicker terminal differentiation. The reviewer is not convinced, please tone down the title.  

Response: Thanks for your suggestion, we have changed the title in the revised manuscript.

Comments:

1/ I don’t understand the role of the following sentence in the text.

“The embryonic development of pineal bolism begins with serotonin synthesis, followed by serotonin acetylation and 5-hydroxyindole methylation in goose (18).” 

Please remove.

Response: Thanks for your suggestion, we have removed this sentence in the revised manuscript.

2/ possitive ln 356 – please correct

Response: Thanks for your suggestion, we have corrected this sentence in the revised manuscript.

3/The paragraph starting at ln 363 is a bit unclear: Please rephrase the paragraph and determine better the classes. : “There was a negative or positive correlation between DNA methylation and gene expression in certain regulatory regions in all stages we studied. According to RNA-seq data, genes were classified into four groups based on their FPKM expression levels (high expression, middle expression, low expression and no expression). CG methylation levels in gene regulatory regions (upstream 2 kb, gene body and downstream 2 kb) were analyzed in each group.“

Response: Thanks for your suggestion, we have revised this paragraph in the submitted manuscript.

4/ What was the minimal number of reads for a CG to be included in the study? Was there any methylation difference required for the identification of a DMR?

Response: The minimal number of reads for a CG to be included in the study is 4.The identification requirements of DMR are as follows: for the CG difference in a certain range, we require that the number of CG in the window should not be less than 5, the absolute value of methylation difference should be ≥ 0.25, and the Q value should be ≤ 0.05. We have added the above contents in the Materials and Methods section of revised manuscript.

5/ Please check the methylation and expression level of the TeT genes. 

Response: Thanks for your suggestion. Tet1 showed significantly higher expression in WZE than in STE at E15, while Tet2 and Tet3 showed significantly higher expression in WZE than in STE at E23. At P1, Tet2 showed significantly higher expression in STE than in WZE. The DNA methylation level of Tet1, Tet2 and Tet3 showed no significant differences between two goose breeds. We have added the new results in Figure S3 and in the result section of the revised manuscript.

6/ Did you observe any methylation difference between Zwe and STE at enhancers, CpG shores and shelves?

Response: In this study, we mainly focus on the DNA methylation differences in the gene regulatory regions (upstream 2 kb, exon, intron, and downstream 2 kb), and carried out correlation analysis with transcriptome data. DNA methylation of enhancers, CpG shores and shelves also play an important role in the regulation of gene expression, but they have not been analyzed in this study. We will make further analysis in future research.

7/ The quality of images is very low, sometimes it’s impossible to read the legends. On figure 5, the green/red P values scale is not explained. Are they corrected P values? The non-significant values should not be indicated with such a strong green color. I’d propose grey. The color-code might rather be changed to methylation difference of the significant DMRs. Please switch to blue-yellow coding.

Response: Thanks for your suggestions. The numbers in the figure indicate different corrected p-values, we have added it in the revised manuscript. The color code of Figure 5 have been changed according to your suggestion. All the figures have been corrected according to the guideline of this journal, and the original figures can be found in the attached files.

8/ On figure 7b the rectangle really corresponds to 300 kb?

Response: We are sorry not making the data accurate enough. On Figure 7B, the rectangle corresponds to 338 kb. We have changed Figure 7B in the revised manuscript.

9/ Figure 7a is not really convincing. The MyoD1 expression kinetic trend determined by the two different methods is similar but quantitatively seems to be different, which might have physiologic relevance. Please repeat the qPCR experiment on new samples.

Response: Thanks for your suggestion, we have repeated the qPCR experiment on new samples. We have revised Figure 7A in the submitted manuscript.

10/ DNA methylation level representation on 7d is not clear. Legend is missing. Please find a representation, which is more obvious. The greyscale doesn’t allow to easily identify the differences.

Response: We are sorry making some part of the figure legend missing. We have added the figure legend and changed the colors of Figure 7D in the revised manuscript.

11/ The discussion is too lengthy. Please shorten. The reviewer considers that the presented results are rather negative, do not really show the role of the DNA methylation in the observed phenotypic difference between the two breads. Please include this in the discussion. Negative results are important!

Response: Thanks for your kind suggestion, we have shortened and revised the discussion section according to your advice. In addition, the markers of significant differences of Figure 6B and 7B were missing in the original submitted manuscript, we have corrected them in the revised manuscript.

Reviewer 2 Report

Dear Authors, in my opinion your manuscript has to be refined for spelling mistakes and for format errors. In particular in the initial list of the authors there is a sentence incorrectly inserted. In the abstract semi-colons were introduced instead of commas, and so on.

I'd make the title more concise and attractive.

In the introduction section I suggest to move the paragraph from lines 79 to 88 at the beginning, adding further information about geese (Wuzong gees and Shitou geese) so implementing it.

Conclusions are not mandatory, but I suggest adding this section.

The quality of presentation is good; the study design is robust (even if the sampling analysed number is low) such as the adopted approaches and techniques that are novel, well detailed, well described and clearly reported.

Proably, in my humble personal point of view, the paper embraces a niche topic so a few researcher readers at this level, but anyway it is interesting.

Please, check that references are in line and in accordance with IJMS required format.

Best regards.

M.T

Author Response

Dear Reviewer,

Thank you for your comments concerning our manuscript entitled "Earlier embryonic leg muscle myogenic gene demethylation contributes to more rapid myoblast terminal differentiation in Wuzong geese than in Shitou geese" (ijms-2266565). Those comments are all valuable and very helpful for revising and improving our paper, as well as the important guiding significance to our researches. We have studied all the comments carefully and modified the paper according to the reviewer’s suggestions. Revised portion are highlighted with tracked changes in the manuscript.

It would be greatly appreciated if our revised manuscript could be considered for publication in International Journal of Molecular Sciences.

Sincerely yours,

Xumeng Zhang

-----------------------------

The following is a point-to-point response to the reviewer’s comments.

Reviewer’s Comments to Author:

Reviewer 2:

Dear Authors, in my opinion your manuscript has to be refined for spelling mistakes and for format errors. In particular in the initial list of the authors there is a sentence incorrectly inserted. In the abstract semi-colons were introduced instead of commas, and so on.

Response: Thanks for your kind suggestions, we have corrected the spelling mistakes and format errors throughout the manuscript.

I'd make the title more concise and attractive.

Response: Thanks for your suggestion, we have revised the manuscript according to your advice.

In the introduction section I suggest to move the paragraph from lines 79 to 88 at the beginning, adding further information about geese (Wuzong gees and Shitou geese) so implementing it.

Response: Thanks for your suggestion, we have revised the introduction section according to your advice.

Conclusions are not mandatory, but I suggest adding this section.

Response: Thanks for your suggestion, we have added conclusion section in the revised manuscript.

The quality of presentation is good; the study design is robust (even if the sampling analysed number is low) such as the adopted approaches and techniques that are novel, well detailed, well described and clearly reported.

Proably, in my humble personal point of view, the paper embraces a niche topic so a few researcher readers at this level, but anyway it is interesting.

Please, check that references are in line and in accordance with IJMS required format.

Response: Thanks very much for your highly praised comments concerning our study, which give us confidence in future study. We have corrected the references and format according to the guideline of IJMS.

Best regards.

M.T

Round 2

Reviewer 1 Report

I appreciate the authors’ answers to my comments, and I consider that the manuscript is now clearly improved.

However, I have some further comments, only in relationship with my previous ones. I maintain therefore the original numbering.

5/ Please confirm the expression changes of the Tet genes by qPCR.

6/ I understand that performing methylation analysis of enhancers of the goose genome might be complicated. However, investigating the methylation differences between CpG islands, shelves, and shores is easy. I ask the authors to perform this analysis and include the results instead of simply saying that they’ll do this in the future. 

7/ The quality of images is still poor. I did not find the separate files. The images included in the manuscript are hardly readable.

On figure 5, non-significant changes are still prominent. I’d propose to put much larger redscale or bluescale for significant results with a single grey tone without scaling for the non-significant results. There’s no use to include p-values for non-significant results.

8/ It is unclear to me that on figure 7b the authors discuss the methylation of a region of 338kb. On panels 7c and 7d they show results about a region of 300bp. How do these two regions relate to each other. Why don’t the authors show this 300bp methylation as well on panel 7b?

9/ Please include the number of qPCR performed for MyoD. Please include the results of the first set of qPCRs included in the previous version of the manuscript.

10/ The color code on 7d is still difficult to identify. Dark blue is almost black. Please don’t use the dark blue, include white for the most hypomethylated cytosines and shift the two other blues.

11/ The end of the discussion is a bit unclear, please reformulate the sentences.   

Round 3

Reviewer 1 Report

Dear Authors,

Thank you for your answers. The manuscript has significantly improved.

Still some minor changes are required:

1/ Supplementary figures have very poor legends. Please prepare detailed legends, indicate n values for parallels.

2/ Figure 7 legend: please include n values for parallels for 7a

3/ The authors still continue to confound kb and bp in their answers and on the figure 7. Finally, I understood the black rectangle corresponds to bp instead of kb. Please correct.

4/ In supplementary table 6 please highlight the 13 CpGs analyzed by pyrosequencing as well.

5/ The last sentences of the discussion are still difficult to understand. Please rewrite them in correct English. Further, please omit the reference to the analysis of CpG islands what you have already carried out.
